# The Link Between Left Atrial Longitudinal Reservoir Strain and Mitral Annulus Geometry in Patients with Dilated Cardiomyopathy

**DOI:** 10.3390/biomedicines13071753

**Published:** 2025-07-17

**Authors:** Despina-Manuela Toader, Alina Paraschiv, Diana Ruxandra Hădăreanu, Maria Iovănescu, Oana Mirea, Andreea Vasile, Alina-Craciun Mirescu

**Affiliations:** 1EuroEchoLab Craiova Cardiology Center, Emergency Hospital Craiova, 200642 Craiova, Romania; oana.mirea83@gmail.com (O.M.); alinameds@yahoo.com (A.-C.M.); 2Doctoral School, University of Medicine and Pharmacy of Craiova, 200349 Craiova, Romania; 3Department of Cardiology, University of Medicine and Pharmacy of Craiova, 200349 Craiova, Romania; diana.hadareanu@umfcv.ro (D.R.H.); maria.iovanescu@umfcv.ro (M.I.)

**Keywords:** dilated cardiomyopathy, functional mitral regurgitation, left atrial reservoir strain, mitral annulus remodeling, four-dimensional echocardiography

## Abstract

**Background/Objectives**: Anatomical and functional damage of the mitral valve (MV) apparatus in patients with dilated cardiomyopathy (DCM) is secondary to left ventricular (LV) injury, leading to functional mitral regurgitation (FMR). Real-time four-dimensional echocardiography (RT 4DE) is a useful imaging technique in different pathologies, including DCM. Left atrial (LA) strain, as measured by left atrium quantification software, is an accurate technique for evaluating increased filling pressure. The MV has a complex three-dimensional morphology and motion. Four-dimensional echocardiography (4DE) has revolutionized clinical imaging of the mitral valve apparatus. This study aims (1) to characterize the mitral annulus (MA) parameters in patients with DCM and advanced-stage heart failure (HF) according to etiology and (2) to find correlations between left atrial function and MA remodeling in this group of patients, using 4DE quantification software. **Methods:** A total of 82 patients with DCM and an LV ejection fraction ≤ 40% were recruited. Conventional 2DE and RT 4DE were conducted in DCM patients with a compensated phase of HF before discharge. The measured parameters were left atrial reservoir strain (LASr), annular area (AA), annular perimeter (AP), anteroposterior diameter (A-Pd), posteromedial to anterolateral diameter (PM-ALd), commissural distance (CD), interregional distance (ITD), annular height (AH), nonplanar angle (NPA), tenting height (TH), tenting area (TA), and tenting volume (TV). **Results:** Measured parameters revealed more advanced damage of LA and MA parameters in ischemic compared to nonischemic etiology. Univariate analysis identified AA, AP, A-Pd, PM-ALd, CD, ITD, TH, TA, and TV (*p* < 0.0001) as determinants of LASr. Including these parameters in a stepwise multivariate logistic regression, PM-ALd (*p* = 0.03), TH (*p* = 0.043), and TV (*p* = 0.0001) were the best predictors of LAsr in these patients. **Conclusions:** The results of this study revealed the correlation between LA function depression and MA remodeling in patients with DCM.

## 1. Introduction

Dilated cardiomyopathy (DCM) is a left ventricular (LV) or biventricular disease characterized by dilation and systolic dysfunction, without abnormal loading conditions such as valvular heart disease, hypertension (HTN), or coronary artery disease, severe enough to explain this dysfunction [1]. DCM represents 70% of heart failure (HF) cases [2]. Diffuse reactive interstitial fibrosis, resulting from increased collagen proliferation by myofibroblasts, characterizes the histopathological features of DCM [3]. Anatomical and functional damage of the mitral valve apparatus (MVA) in patients with DCM is secondary to LV injury, leading to functional mitral regurgitation (FMR) and creating a vicious circle in which FMR maintains and aggravates itself [4]. This disease has a progressive evolution with an unfavorable prognosis in most cases. The primary imaging technique for diagnosing and evaluating DCM is two-dimensional transthoracic echocardiography (2D TTE) [5,6]. The myocardial walls have a complex feature, and standard echocardiographic parameters used for systolic and diastolic function are insufficient to assess cardiac performance [7].

Mitral annulus (MA) morphology and dynamics are abnormal in patients with DCM and FMR [8] and are linked to diastolic ventricular filling [9]. Using more sensitive echocardiographic techniques may enhance the understanding of the pathophysiological mechanisms underlying the evolution of this disease. Real-time four-dimensional echocardiographic (RT4DE) imaging techniques offer valuable insights in various clinical settings, including dilated cardiomyopathy (DCM). One notable method is the measurement of left atrial strain during the reservoir period (LASr) using four-dimensional auto left atrium quantification (4D Auto LAQ), which serves as an accurate tool for assessing increased filling pressure [10]. The mitral valve (MV) has a complex three-dimensional structure and movement. Advances in RT4DE have transformed the clinical imaging of the mitral valve, enabling realistic visualization. The 4D mitral valve quantification (4D Auto MVQ) technique allows for precise segmentation and quantification of the intricate morphology and motion of the mitral valve apparatus (MVA). The quantitative analysis offered by RT4DE has deepened our understanding of the pathological mechanisms underlying degenerative, ischemic, functional, and rheumatic mitral valve diseases [11]. In patients with DCM, both LASr and mitral valve apparatus geometry are often abnormal, indicating advanced chamber remodeling. Evaluating these parameters may help in monitoring therapeutic responses and guiding interventions for functional mitral regurgitation (FMR) [11].

This study aimed to (1) characterize the features of LASr and mitral valve parameters in patients with DCM at an advanced stage of heart failure, categorized by etiology, and (2) identify correlations between left atrial (LA) function and mitral valve remodeling in this patient group using 4D echocardiographic quantification. 

## 2. Materials and Methods

### 2.1. Study Population and Design

This study is an observational retrospective analysis conducted in a hospital setting, focusing on patients diagnosed with DCM who were admitted due to signs and symptoms of decompensated heart failure (HF). Patients were sent for a comprehensive echocardiographic evaluation. A 2D TTE definitively confirmed the diagnosis of DCM in line with the latest clinical guidelines. This important research involved 97 DCM patients who experienced heart failure decompensation and were admitted between January 2020 and December 2022 at the Department of Cardiology of the Emergency County Hospital of Craiova, highlighting the critical care provided during this challenging period.

The participants were aged 31 to 80 years (mean age 60 ± 10). The echocardiographic images were acquired before discharge in a stable, compensated condition, using a commercial Vivid E9 ultrasound system (GE Vingmed Ultrasound, Horten, Norway) with an M5S probe (frequency: 1.5–4.6 MHz). The analysis was performed offline using EchoPAC version BT204 dedicated software. (GE Vingmed Ultrasound Systems, Horten, Norway). The following biochemical tests were conducted upon admission and throughout hospitalization: hemoglobin, glycated hemoglobin, urea, creatinine, potassium, sodium, LDL, HDL cholesterol, and triglycerides. The Modification of Diet in Renal Disease (MDRD) equation was used to calculate the glomerular filtration rate [12]. The following criteria were strictly enforced for exclusion: cardiogenic shock, hypervolemia, severe malignant arrhythmias, advanced atrioventricular block, uncontrolled heart rates that would compromise proper echocardiographic acquisition, suboptimal echocardiographic windows, and any patients who refused to participate. The exclusion criteria were clearly defined to ensure the safety and integrity of the study. Patients with cardiogenic shock, characterized by the heart’s inability to supply adequate blood flow, were not eligible. Those with hypervolemia, an excessive fluid volume in the circulatory system, and individuals experiencing severe malignant arrhythmias, which can lead to life-threatening heart rhythms, were also excluded. Additionally, patients with advanced atrioventricular block, a condition that disrupts the electrical signals between the heart’s chambers, and those with uncontrolled heart rates that could hinder effective echocardiographic imaging were excluded from participation. Finally, any individuals who opted out of the study were also excluded from consideration. Twenty-three patients presented exclusion criteria, and 8 patients were excluded due to inadequate transthoracic echocardiography for 4D TTE acquisition. The final cohort consisted of 82 DCM patients, divided into two groups: ischemic (38 patients) and nonischemic (44 patients). The evidence for ischemic etiology is supported by the presence of coronary artery disease, as confirmed by angiography showing luminal narrowing of 70% or more in multiple vessels. Additionally, a history of percutaneous interventions (PCI) or coronary artery bypass grafting (CABG) further reinforces this diagnosis. Heart rate, blood pressure, height, and weight were measured before the echocardiographic image acquisition. The body surface area BSA was calculated using the DuBois formula [13]. The patients received medical treatment according to current guidelines. All study participants gave written informed consent. The study was compiled with the Declaration of Helsinki, approved by the Local Committees for Medical and Health Research Ethics (no. 74 from 7.09.2020), and conducted under the local institutional requirements.

### 2.2. Echocardiography

#### 2.2.1. Two-Dimensional Echocardiography

Two-dimensional TTE images were acquired in all patients before discharge, after careful compensation for hydrostatic retention, to determine the cardiac chamber dimensions and function according to the current ASE/EACVI recommendations [7]. The results confirmed the DCM diagnosis [7]. Standard M-mode, two-dimensional, color, pulsed, and continuous-wave Doppler images were acquired. Echocardiographic images were captured using standard views, carefully avoiding foreshortening and strictly following the chamber quantification guidelines to ensure accuracy and reliability [7]. To ensure the integrity of the results, subjects with poor image quality—characterized by more than three poorly delimited segments—were excluded from the study. The LV apical views—4-chamber (A4ch), 3-chamber (A3ch), and 2-chamber (A2ch)—along with long-axis and short-axis views, were meticulously acquired with the patient positioned in the left lateral decubitus posture during a breath hold. Careful adjustments of the gain and compression settings ensured that the LV endocardial and epicardial borders were clearly visualized, minimizing any dropout. Additionally, the depth and sector angle were optimized to capture the entire LV while maintaining a frame rate of 40 to 60 frames per second, enhancing image quality. The LV ejection fraction (LVEF) was precisely calculated using the biplane Simpson method, adhering to current guidelines, with a critical cut-off value for study inclusion set at <40%. The left atrium (LA) volume was computed via the modified Simpson’s rule and indexed to body surface area (BSA), providing a comprehensive assessment. Valvular functionality was thoroughly examined using advanced imaging techniques, including two-dimensional, color, pulsed, and continuous-wave Doppler echocardiography [14]. For accuracy, measurements were averaged across three cardiac cycles for patients in sinus rhythm (SR) and five cycles for those presenting with atrial fibrillation (AF), ensuring robust and reliable data for clinical evaluation [14].

The assessment of left atrial (LA) volume was conducted using both A4-chamber (A4ch) and A2-chamber (A2ch) echocardiographic views, obtained to ensure the highest accuracy in quantification. To effectively prevent foreshortening of the LA, maximized visual representation of both the length and base within each view was used. The disparity in LA lengths between the A4ch and A2ch views remained below 10% [15], reinforcing the reliability and consistency of the measurements. For the purpose of volume analysis, the biplane disk summation method was employed at the end-systole, just before the MV opening, providing a precise measurement point. It was essential to exclude the LA appendage and the ostia of the pulmonary veins from calculations to enhance the accuracy of the LA volume assessment. Diastolic mitral peak velocities, including both the early (E) and late (A) phases, as well as the E-wave deceleration time (DT) were measured. These parameters were obtained using pulsed-wave Doppler echocardiography, allowing the capture of dynamic changes in blood flow. Furthermore, peak myocardial diastolic velocities, represented as early (e′) and late (a′), were meticulously measured using tissue Doppler at both the medial and lateral annulus levels from the apical 4-chamber view. The average of these values was calculated to provide a clearer image of myocardial function. The E/e′ ratio was computed, offering critical insights into LA dynamics and overall cardiac performance [15].

Patients with satisfactory image quality underwent a 4D echocardiographic examination using a 4V transducer. The frame rate was set to 40% of the heart rate.

#### 2.2.2. Four-Dimensional Transthoracic Echocardiography

Four-dimensional echocardiograms were acquired using live 4D imaging in AF patients and electrocardiographically (ECG)-gated 4DE acquisition for SR patients. In live 4D imaging, a volumetric data set of a relatively narrow pyramidal sector is acquired and displayed in RT. In ECG, multiple-beat acquisition volumes are stitched together during four consecutive heart cycles during a single-breath hold. The minimum volume rate for LA 4DE strain measurements was 25 volumes per second (VPS) (Figure 1A); for MV 4DE measurements, it was 16 volumes per second (Figure 1B). Care had been taken regarding the volume rate because, as previously shown, a low temporal resolution might lead to an underestimation of strain magnitude [16,17,18].

A full-volume image in the A4ch view was acquired and stored. For the accuracy of the measurements, efforts were made to collect high-quality images. Patients with inadequate images for analysis were excluded from the study. Images with inadequate visualization of the LA or artifacts were excluded from the analysis. Four-dimensional analysis was performed using commercially available EchoPAC version BT204 dedicated software (GE Vingmed Ultrasound Systems, Horten, Norway) with specifically designed 4D Auto LAQ and 4D Auto MVQ.

For the LA quantification, the landmark was located in the center of the MV at the annulus level in the three planes (Figure 2). The segmentation algorithm computes the deformation of the model. For LAS longitudinal and circumferential strain evaluation, the ‘review’ function was selected. The software automatically detected the endocardial border in three-dimensional space throughout the cardiac cycle. Care was taken to ensure correct delineation by rotating along the long axis of the LA. The investigator manually adjusted the border at end-diastole (Figure 2A), end-systole (Figure 2B), and pre-atrial contraction (Figure 2C) to include the entire atrium and exclude the pulmonary veins and LA appendage. The software automatically generated all left atrium parameters, including the LAS (Figure 3) [19].

The definition of the LA cycle was R-R gating, in which the LA cycle is defined from the peak of the R-wave to the same point of the next cycle. The method presented the LA strain as a monophasic curve (Figure 3). The advantage of the R-R gating method is its availability even during AF [20]. The reference point was set at the left ventricle ED. During the reservoir phase, the LA wall lengthened, and the strain had a positive value. In the other 2 phases, the LA wall shortened, and the strains had negative values. LA longitudinal strain parameters obtained using this technique included LA reservoir strain (LASr), LA conduit strain (LAScd), and LA contraction strain (LASct), and LA circumferential strain parameters included the LA reservoir circumferential strain (LASr-c), LA conduit circumferential strain (LAScd-c), and LA contraction circumferential strain (LASct-c). The studies showed that the total function of the LA was best reflected by reservoir strain [21,22]. From all the 4D LA parameters presented by the software, this parameter was used for the LA function evaluation.

Four-dimensional echocardiography image acquisition of the MV for quantitative analysis is usually performed in the A4ch view on transthoracic echocardiography [23,24]. An optimal view was selected for the most MV favorable segmentation. The 4D Auto MVQ software package enabled the detection of anatomic landmarks (Figure 4), followed by surface modeling using a geometric mesh. Finally, we obtained the annular and leaflet geometry, as well as their dynamics (Figure 5) [25,26,27].

The IBM SPSS Statistics version 26.0.1.1. was used for statistical analysis. Baseline characteristics and echocardiographic parameters were defined as mean ± standard deviation (SD) for continuous variables and absolute number (n) or percentage for categorical variables. The normality or skewness of the variables was checked using the Kolmogorov–Smirnov test. To compare data between the two pre-defined groups, an unpaired *t*-test was used. Pearson’s correlation was analyzed between pairs of continuous variables. Binary logistic regression (simple and multivariate) analysis identified echocardiographic parameters associated with LASr. Variables with statistically significant univariate regression were used for multivariate stepwise regression analysis. Significance was defined as a two-tailed probability level of *p* < 0.05 for all tests.

A single experienced sonographer conducted both the acquisition of echocardiographic images using a 4V probe (frequency: 1.5–4.0 MHz) and the subsequent image postprocessing. To assess intra-observer reliability, the sonographer re-analyzed images from 10 randomly selected patients 30 days after the initial analysis. Intra-observer variability was evaluated by calculating the intra-class correlation coefficient (ICC) and the 95% confidence intervals (CIs) for the strain components.

## 3. Results

### 3.1. Basic Characteristics Are Summarized in Table 1

The ischemic group consisted of 38 patients (46%), with a mean age of 60 years. The patients in the nonischemic group were statistically significantly younger than those in the ischemic group (57 ± 12 vs. 63 ± 8 years, *p* = 0.016). A total of 62% of the subjects were men (60% in the ischemic group and 64% in the nonischemic group). The two groups had similar BSA, blood pressure, and heart rate (Table 1). Cardiovascular risk factors were more prevalent in the ischemic group, as follows: arterial hypertension (HTN), 79% vs. 52%, *p* = 0.012; diabetes mellitus (DM), 61% vs. 36%, *p* = 0.03, and dyslipidemia, 63% vs. 48%, *p* = 0.004. A higher proportion of patients with chronic kidney disease (CKD) were found in the ischemic group compared with the nonischemic group, 47% vs. 27%, *p* = 0.06. The prevalence of atrial fibrillation (AF) was 34% in the ischemic group and 27% in the nonischemic group, *p* = 0.5. A total of 10 subjects (31%) died, and 17 (45%) were readmitted from the ischemic group, compared with 15 subjects (34%) who died and 22 (50%) who were readmitted in the nonischemic group.

### 3.2. 2D TTE Results

LV end-diastolic volume (LVEDV) provided lower values of 200.97 mL (SD 71.13) in NIDCM subjects and 203.86 mL (SD 82.76) in IDCM subjects (*p* = 0.86); LV end-systolic volume (LVESV) was higher in patients with NIDCM, 149.61 mL (SD 63.65) vs. 149.55 mL (SD 69.54) in patients with IDCM (*p* = 0.99). LVEF was 27.45% (SD 7.7) in the IDCM group and 26.52% (SD 7.82) in the NIDCM group (*p* = 0.59) (Table 2). 

Left atrium volume indexed (LAVi) was 54.23 mL/m^2^ (SD 12.6) in the case of ischemic etiology compared with nonischemic etiology, 54.03 mL/m^2^ (SD 14.84) (*p* = 0.95) (Table 2). The number of patients with MR ≥ mild and TR ≥ was higher in the NI group (17 vs. 16 (*p* = 0.75) and 17 vs. 10 (*p* = 0.24), respectively).

### 3.3. 4D TTE Results

Four-dimensional LAQ provided lower values of LASr in ischemic patients compared with nonischemic patients (8.44% (SD 3.68) vs. 8.5% (SD 4.42)) but without significant statistical differences between the two groups (*p* = 0.94) (Table 3).

Four-dimensional MVQ analysis revealed higher values of MA parameters in the ischemic vs. nonischemic group, indicating more advanced damage, as follows: AA 20.81 cm^2^ (SD 3.91) vs. 19.79 cm^2^ (SD 2.71) (*p* = 0.17), AP 16.44 cm^2^ (SD 2.05) vs. 16.35 cm^2^ (SD 2.09) (*p* = 0.84), A-P diameter 4.61 cm (SD 0.47) vs. 4.56 cm (SD 0.65) (*p* = 0.65), PM-AL diameter 5.27 cm (SD 0.72) vs. 5.1 cm (SD 0.58) (*p* = 0.24), CD 4.96 cm (SD 0.64) vs. 4.94 cm (SD 0.52) (*p* = 0.81), ITD 3.73 cm (SD 0.78) vs. 3.67 cm (SD 0.63) (*p* = 0.63), AH 6.86 cm (SD 2.21) vs. 6.5 cm (SD 2.06) (*p* = 0.44), and NPA 156 degrees (SD 15.7) vs. 155.77 degrees (SD 13.91) (*p* = 0.83). Tenting parameters were also higher in IDCM compared with NIDCM patients, as follows: TH 1.51 cm (SD 0.31) vs. 1.48 cm (SD 0.34) (*p* = 0.68), TA 4.65 cm^2^ (SD 1.25) vs. 4.59 cm^2^ (SD 1.53) (*p* = 0.84), and TV 13.31 cm^3^ (SD 5.32) vs. 11.92 cm^3^ (SD 5.6) (*p* = 0.27) (Table 3).

Using the Pearson method, we found a significant correlation between MA parameters and LASr, except for AH and NPA, both in the ischemic and nonischemic groups (Table 4) (Figure 6 and Figure 7). Univariate analysis identified AA, AP, AP diameter, PM-AL diameter, CD, Itd, TS, TH, and TV (*p* < 0.0001) as determinants of LASr (Table 5). When including these parameters in a stepwise multivariate logistic regression, PM-AL diameter (*p* = 0.03), TH (*p* = 0.043), and TV (*p* = 0.0001) were the best predictors of LASr in patients with DCM and an advanced stage of HF (Table 6) (Figure 8).

### 3.4. Reproducibility of the Measurements

The intra-observer reproducibility was excellent for 4D TTE LA and MA parameters. Repeat 4D Auto LAQ and 4D Auto MVQ measurements of 15 randomly selected participants were taken by the same observer two weeks after the first measurement, which allowed for the determination of intra-class correlation coefficients (ICC) to measure intra-observer agreement. The ICC values ranged from 0.861 to 0.999 (Table 7), indicating good intra-observer reproducibility. Table 8 provides cross-validation procedure results. 

## 4. Discussions

The present study evaluated LASr and MA parameters in DCM patients using RT4DE. The main results were as follows: (1) LASr measurement by 4D Auto LAQ identified lower values of LASr in ischemic patients compared with nonischemic patients, suggesting a more advanced LA dysfunction; however, this difference was not significant. (2) Four-dimensional Auto MVQ analysis also revealed higher values of MA parameters in the ischemic vs. nonischemic groups, indicating more advanced damage, but without significant difference between the two groups, suggesting that, in DCM in the advanced stage of the disease, when the LV is severely remodeled and LVEF is blunted, the hemodynamic factors become more important than risk factors [28]. (3) Multivariate logistic regression identified that PM-AL diameter, TH, and TV were the best predictors of LASr in this patient group.

*The mitral valve apparatus (MVA)* is an asymmetrical dynamic structure consisting of valvular leaflets, MA, and a sub-valvular apparatus with tendinous chords and papillary muscles [9,29,30]. MA has a three-dimensional saddle shape, with “peaks” at the level of the midpoints of the anterior and posterior leaflets and “valleys” at the commissural level [31,32]. Anatomists define MA as a straight anterior and a C-shaped posterior segment with different structures and functions [33]. The atrial myocardium is an integral part of the posterior MA and is important in the “pre-systolic” phase [34,35]. The MA posterior segment results from the convergence of the following four structures: the atrial wall, the leaflet hinge line, the marginal free wall of the LV, and adipose tissue [34]. During the LV contraction, the MA area decreases by 20–30%. The smallest MA area occurs during isovolumetric contraction, and the largest area occurs during isovolumetric relaxation [36].

In healthy humans, the MA is essential for the structural and functional integrity of the MV complex, reducing stress exerted by LV forces on the mitral valve leaflets [32,37]. During the LV contraction, the MA folds from early to mid-systole and increases its saddle-shape configuration. During late systole to diastole, the MA gradually expands back [38]. Four-dimensional echocardiography has significantly enhanced the understanding of the anatomy of the mitral valve apparatus [39,40,41]. RT4DE is a useful imaging technique for identifying differences in MA shape and function in cardiac diseases [8]. MA morphological parameters provided by RT4DE are AA, AP, APd, PM-ALd, CD, ITD, AH, NPA, and MAA [38,42] (Figure 5). 

*Annular characteristics are abnormal in patients with DCM and FMR.* Chronic LV remodeling produces enlarged, flat, and adynamic [43,44,45] MA, with apical displacement of papillary muscles (PMs) [46] (Figure 9). The MA contraction at early systole is decreased or absent [45,47,48,49,50,51]. Studies have shown that the attenuated contractility of MA in FMR is significantly associated with a loss of physiological morphology [52,53,54,55,56]. MA dilation in DCM does not comprise the entire annular circumference. The most damaged is the posterior segment. The dilation is asymmetric, increasing septal–lateral diameter, impeding the leaflet’s coaptation, and resulting in FMR [34].

The anatomical features of the MVA in patients with DCM and FMR can be described as follows: increased distance between the MA and interregional zone, increased annular nonplanarity scalar angle, and increased anteroposterior and posteromedial–anterolateral diameters, with loss of the typical saddle shape of the MA [49,50,51]. The evaluation of MA geometry and function, as well as the determinants of MA remodeling, is essential for understanding the pathophysiology and severity of FMR in patients with DCM, particularly in the context of reparative surgery [57,58].

Four-dimensional MA parameters provided by 4D Auto MVQ in the present study revealed more severe damage in IDCM compared with NIDCM patients, although the differences were not statistically significant. The results were in accordance with revealing > mild FMR more frequent in ischemic etiology, indicating more advanced remodeling and dysfunction in this group of patients. The results identified comparable values of echocardiographic parameters, suggesting that, in the advanced stage of the disease, MA presents comparable quantitative morphological damage in both ischemic and nonischemic etiologies of cardiac dilatation.

The connection between MA remodeling and left heart chamber function was not defined. The influence of atrial function on annular shape has not been well studied in previous studies. The research of Noacka T. et al. showed the effect of atrial contraction on MA [42]. Patients with MR showed dilatation in both the anteroposterior and lateromedial directions, resulting in a rounder shape of the MA compared to the control groups. In normal subjects, the MA exhibits a twin-peaked curve, which is explained by early mitral inflow (first peak, E wave) and atrial contraction (second peak, A wave). This research found late-diastolic changes of MA dimensions corresponding to atrial contraction in patients with MR [41,43]. The twin-peaked course of MA was abrogated in MR compared to the control subjects, explained by the higher prevalence of atrial fibrillation, with loss of atrial contraction. The results of this study sustained the effects of atrial function on MA contraction [43].

The present study also identified a correlation between LA function, as evaluated by LASr, and MA parameters.

Previous research has shown that LV global longitudinal strain (LV GLS) has become a well-established parameter for assessing LV function [59]. It is also a more sensitive technique for atrial and myocardial performance evaluation in DCM patients [60].

### 4.1. Left Atrial Reservoir Strain 

Elevated left ventricular (LV) filling pressure is essential for the diagnosis of heart failure (HF) [61]. A comprehensive evaluation of LA volume and function by 2DE and 2D STE is well-studied and associated with important clinical endpoints [62,63,64].

Compared to other echocardiographic techniques, such as tissue Doppler, LA strain is less dependent on angle and load, thereby distinguishing between the active and passive motions of myocardial tissue. It has better repeatability and feasibility [62]. LA strain is also less load-dependent than LAVi [65,66] and orrelated with fibrosis measured by MRI and biopsy [67,68,69]. LA strain evaluated by speckle tracking echocardiography (STE) proved its superiority over conventional echocardiographic parameters for the diagnostic and prognostic evaluation of HF with both reduced and preserved ejection fraction (EF) [70,71,72]. The accuracy of LA strain to identify elevated LV filling pressure is best in patients with reduced LV systolic function [73]. LA enlargement, the decrease in LA pump function, the increase in LV filling pressure, and LV dysfunction all impact the LA reservoir function [73].

The strain curve provides information about LA physiology (Figure 10). LA activity consists of the following three phases in normal conditions: the reservoir, conduit, and contraction phases [74]. The LA reservoir period follows LV diastole. The LA stretches and fills with blood from the pulmonary veins. During this phase, the strain-positive curve increases, reaching the peak at the LV end-systole, before the MV opening (Figure 10, a). The LASr corresponds to LV isovolumic contraction and isovolumic relaxation. After the MV opens, the LA empties quickly until its pressure equals that of the LV. This phase represents the conduit period. During this phase, the strain decreases (Figure 10, b). After the conduit phase, the LA contracts, expelling blood into the LV, and the strain further decreases; [74] this phase represents the LA contraction period (Figure 10, c) [75,76].

Four-dimensional echocardiography for LA evaluation offers prognostic and clinical benefits over traditional 2DE analysis [77], overcoming the limitations of 2D analysis by avoiding geometric assumptions about the complex LA shape [78]. Four-dimensional Auto LAQ software analyzes the structure and function of the LA [74], allowing for longitudinal and circumferential LAS quantification during the cardiac cycle (Figure 3) [79]. In the study by Yafasov, M., the median and limits of normality for LA reservoir strain (LASr) were 30.8% (18.4–44.2%) [80]. In this research, the authors used a Vivid E9 (GE Healthcare, Horten, Norway) for image acquisition and an EchoPAC version 204 for post-processing.

### 4.2. LA Strain in Cardiovascular Diseases

LAS provides pathophysiological information and helps to predict clinical prognosis in various cardiovascular diseases. The study of Chen L. et al. revealed higher values of LASr, LA reservoir circumferential strain, LA conduit circumferential strain, LA contraction strain, and LA contraction circumferential strain in patients with a low risk of thromboembolism compared to those with a high risk of thromboembolism [81]. Abnormal hypertrophy of cardiomyocytes, interstitial hyperplasia, and fibrosis in various cardiac diseases can cause a thickening of the ventricular wall and a decrease in LV and LA compliance, with a decrease in LASr, LAScd, and LASct [82,83,84]. Kuppahally et al. found that LA strain and strain rate, as measured by STE, were significantly correlated with atrial fibrosis [67]. Several studies [85,86,87] have shown that LA strain progressively decreased in all stages of diastolic dysfunction, and LASr could differentiate between diastolic dysfunction stages. Keles et al. [88] found that patients with premature ventricular contraction showed a significant decrease in the LA strain compared to those in the control group, due to atrial and ventricular remodeling, and 4D Auto LAQ can quantitatively detect these changes at an early stage. LASr correlates with pulmonary capillary wedge pressure and is a strong prognostic indicator [89].

LASr and LASct, as measured by 4DE, were independently associated with incident AF and provided incremental prognostic information beyond existing risk scores [81]. LASr was also a significant determinant of death, hospitalization, and new onset AF in HF patients [90]. In patients with coronary disease, hypertensive heart disease, and dilated and hypertrophic cardiomyopathy, the 3D echocardiography-derived LASr showed significant associations with adverse cardiovascular events during the follow-up [91]. Low values of LASr predict poor prognosis among end-stage renal disease patients with preserved LVEF [92].

Previous studies have demonstrated that LA parameters calculated by 4D STE better predict future major cardiac events than those calculated by 2D STE [91,93]. According to previous studies [94,95,96], LA strain is a more accurate and sensitive metric for describing LA function in people with cardiomyopathies, advanced-stage chronic kidney disease, or atrial fibrillation. LA longitudinal strain was also proved to have better predictive performance than other echocardiographic parameters such as LAVi and LAEF [97,98].

LAS evaluated by STE was the strongest predictor of invasively assessed LA fibrosis in patients with end-stage HF [69]. Sun et al. found LASr by 3DE as an independent composite outcome of significant adverse cardiovascular events, including AF in patients without cardiovascular disease undergoing hemodialysis [99].

### 4.3. LA Strain in DCM Patients

LA function is impaired in patients with DCM, reflecting the severity of both diastolic and systolic LV dysfunction [82]. The correlation between the decrease in LA reservoir and pump strain and LV filling pressure is stronger in patients with EF < 50% [73]. Chronic increased LA pressures lead to LA remodeling, characterized by LA enlargement and dysfunction caused by atrial interstitial fibrosis [100]. The substrate of LA remodeling consists of cardiomyocyte atrophy and replacement by fibrotic tissue, causing the thinning of the LA wall [101,102]. This remodeling alters the LA reservoir, conduit, and contractile function, resulting in the loss of systolic function in patients with AF. LA reservoir function is determined by both LV systolic and diastolic function. The studies identified LV filling pressure, LV GLS, and LA volume as independent determinants of LASr [73].

The present study found that, in patients with DCM at the advanced stage of the disease, there are no significant differences between ischemic and nonischemic etiologies regarding LASr values. The explanation lies in the fact that, in the early stages of HF, LASr is influenced by systemic conditions such as HTN [103] DM, DM [104], and dyslipidemia [105]. These risk factors promote LA and LV stiffness via HF fibrosis, hypertrophy, or metabolic stress. In advanced HF, LASr is already severely impaired. Risk factors no longer drive LASr decline as hemodynamic burden and structural damage do. At this stage of the disease, LASr correlates more strongly with the NYHA class of HF, NT-proBNP, left ventricular filling pressures, and left atrial volume than with hypertension, diabetes mellitus, or lipid profile [103,104,105].

LV and LA are anatomically connected, and longitudinal shortening of the LV produces stretching of the LA. In consequence, LV GLS is the strongest determinant of the LASr. Patients with HF and LV systolic dysfunction have increased atrial stiffness [73]. A stiff LA has a low lengthening with a low strain [69]. As shown in the present study, the association between LASr and LV filling pressure depends on the degree of LV systolic dysfunction [73].

The present study employed Pearson correlation analysis and found that AA, AP, PM-AL diameter, CD, Itd, TH, TA, and TV correlated with LASr in patients with DCM. The link between LASr and MA remodeling, identified by this study, is supported by the pathophysiological changes in DCM patients. The LV coordinates the dilation of the LA through the motion of the MA. Therefore, the determining factors of the LA reservoir function are LA stiffness, MA movement, and end-systolic LV volume [74]. During ventricular systole, LA acts as a reservoir. A compliant LA (with a normal strain) expands effectively and “pulls” the MA superiorly and outward. It also contributes to maintaining the MA dynamic saddle shape and area variation. This movement is crucial for reducing leaflet stress and enhancing MV competency. Atrial myopathy, fibrosis, and elevated filling pressures in DCM patients are associated with a reduced LA reservoir strain. All these factors result in blunted annular dynamics, especially decreased systolic annular expansion. They also contribute to annular flattening, reduced area variation, and exacerbate FMR [70].

LA is a component of the posterior MA. Patients with DCM have damaged LA function, as an expression of chronically increased LV filling pressure. Consequently, the effect of LA contractility on MA is decreased in this group of patients.

Using univariate logistic regression, this research revealed that all MA parameters were correlated with LA dysfunction, as evaluated by LASr. By stepwise logistic regression, PM-AL diameter and tenting parameters, TH, and TV correlated with LASr.

This is the first 4D TTE echocardiographic study to characterize MA components and LASr in DCM patients and to investigate a correlation between LASr and MA remodeling. The results are important because, despite the use of advanced imaging tools to evaluate FMR in DCM, these patients have a poor prognosis. In the years to come, the next goal will be to further improve the prognosis of patients with DCM in advanced stages of HF. Knowledge of the MA parameters associated with LA dysfunction may help in developing adequate treatment strategies. Therefore, these findings can be applied to clinical practice in the potential use of a risk score for evaluating disease severity or stratifying patients before surgical planning.

## 5. Conclusions

Combining LASr and MVA assessment using 4D TEE imaging software can provide information about LA function and MA remodeling in DCM patients. This study revealed impaired LA function in patients with DCM and reduced EF. MA dilatation involved the septo-lateral direction predominantly. Multivariate logistic regression identified PM-AL diameter, TH, and TV as predictors of LASr in patients with advanced-stage DCM and HF.

The results of this study suggested a link between LA function depression and MA remodeling in DCM patients, suggesting an interdependence between LA and MA damage, as part of the same process. The clinical implication of these findings is that reduced LASr due to atrial remodeling and fibrosis develops MA dilation and flattening, with resultant FMR in this group of patients. Further studies with a larger number of patients are needed to confirm these results.

### Study Limitation

First, this was a single-center study. Secondly, the study had a relatively small number of patients, and the results should be verified in a larger cohort. The study did not focus on DCM etiology, except for coronary disease. Thirdly, the analysis was performed with a single vendor-dependent software (Echo Pac v204); thus, the results might be vendor-specific. Fourth, the image acquisition and post-processing were performed by a single sonographer. Fifth, this was a retrospective analysis, which may introduce some selection bias.

## Figures and Tables

**Figure 1 biomedicines-13-01753-f001:**
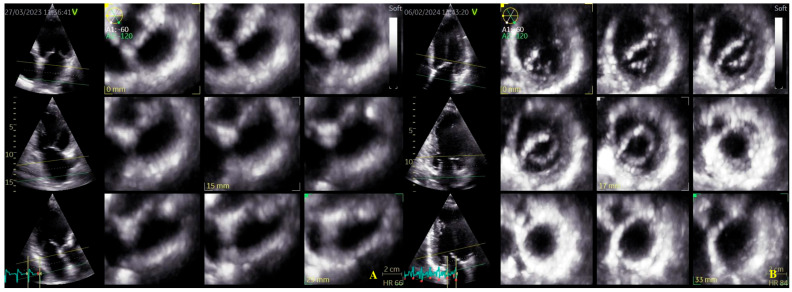
Four-dimensional transthoracic echocardiography (4D TTE) acquisition: (**A**) LA multi-slice acquisition; (**B**) MA multi-slice acquisition.

**Figure 2 biomedicines-13-01753-f002:**
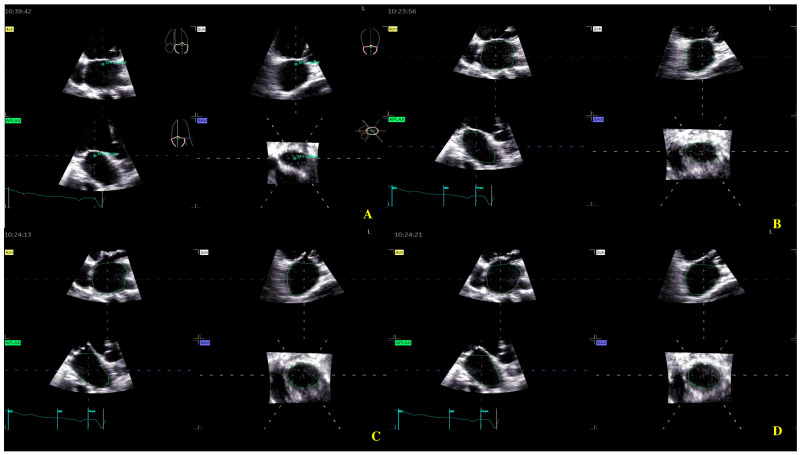
Left atrium segmentation: (**A**) MV center identification; (**B**) left atrium endocardial border in three-dimensional spaces during diastole; (**C**) left atrium endocardial border in three-dimensional spaces during systole; (**D**) left atrium endocardial border in three-dimensional spaces before atrial contraction.

**Figure 3 biomedicines-13-01753-f003:**
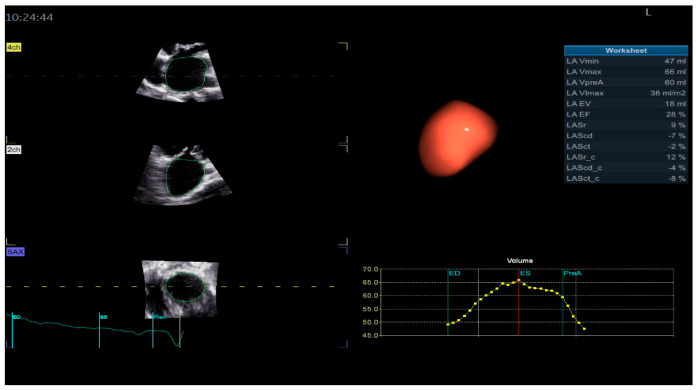
Left atrium parameters provided by Four-Dimensional Auto Left Atrial Quantification (4D Auto LAQ) software, including left atrium reservoir strain.

**Figure 4 biomedicines-13-01753-f004:**
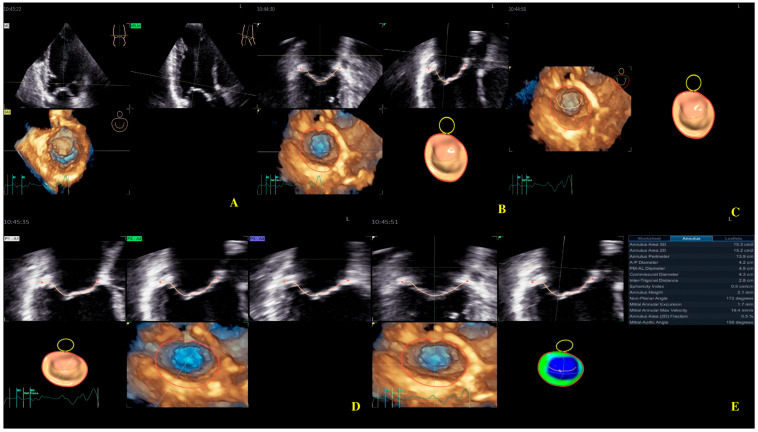
Mitral valve segmentation and quantification (**A**), detection of mitral valve anatomic landmarks (**B**), mitral valve commissures pointing (**C**), mitral valve scallops pointing (**D**), mitral annulus parameters (**E**).

**Figure 5 biomedicines-13-01753-f005:**
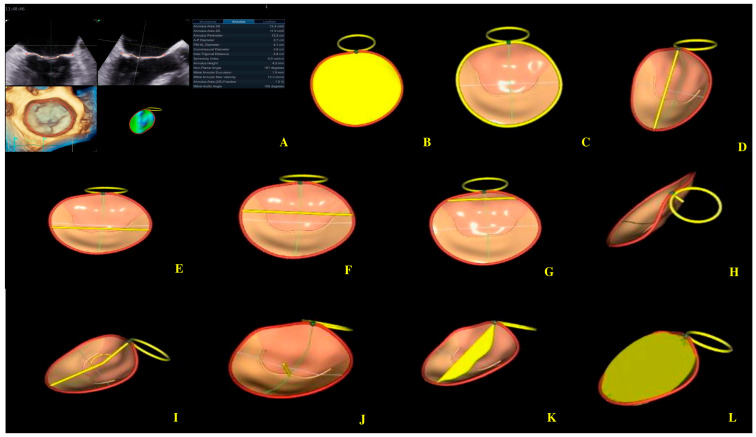
Mitral annulus parameters provided by Four-Dimensional Auto Mitral Valve Quantification (4D Auto MVQ) software: (**A**) Final parameters values; (**B**) AA—annular area; (**C**) AP—annular perimeter; (**D**) AP diameter—anteroposterior diameter; (**E**) PM-AL diameter—posteromedial to anterolateral diameter; (**F**) CD—commissural distance; (**G**) Itd—intertrigonal distance; (**H**) AH—annular height; (**I**) NPA—nonplanar angle; (**J**) TH—tenting heigh; (**K**) TA—tenting area; (**L**) TV—tenting volume.

**Figure 6 biomedicines-13-01753-f006:**
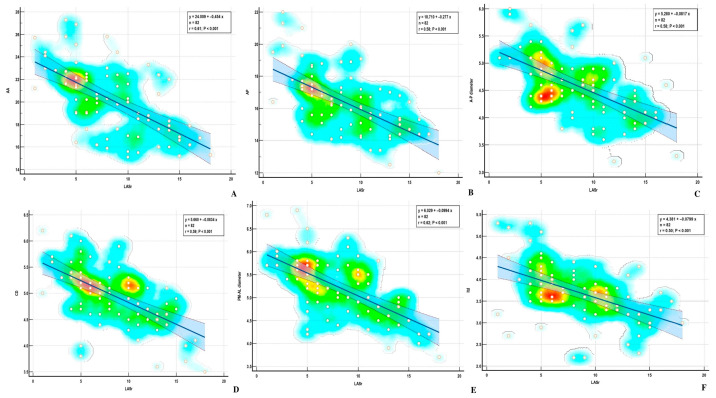
Correlations between LASr and (**A**) AA—annular area; (**B**) AP—annular perimeter; (**C**) A-P diameter—Anteroposterior diameter; (**D**) CD—commissural distance; (**E**) PM-AL diameter—posteromedial to anterolateral diameter; (**F**) Itd—intertrigonal distance.

**Figure 7 biomedicines-13-01753-f007:**
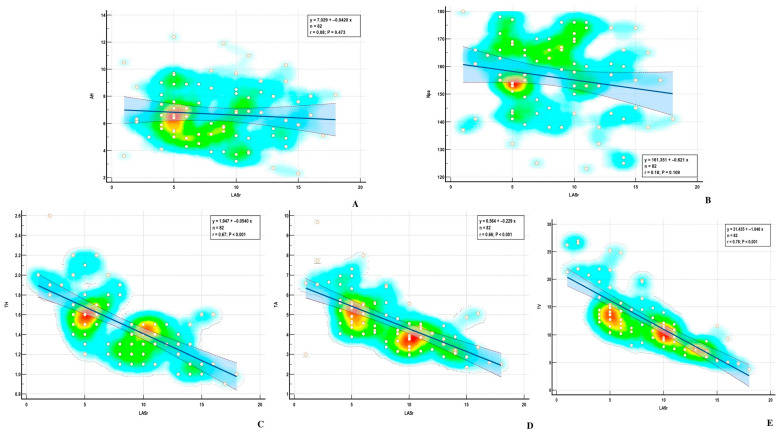
Correlations between LASr and (**A**) AH—annular height; (**B**) NPA—nonplanar angle; (**C**) TH—tenting height; (**D**) TA—tenting area; (**E**) TV—tenting volume.

**Figure 8 biomedicines-13-01753-f008:**
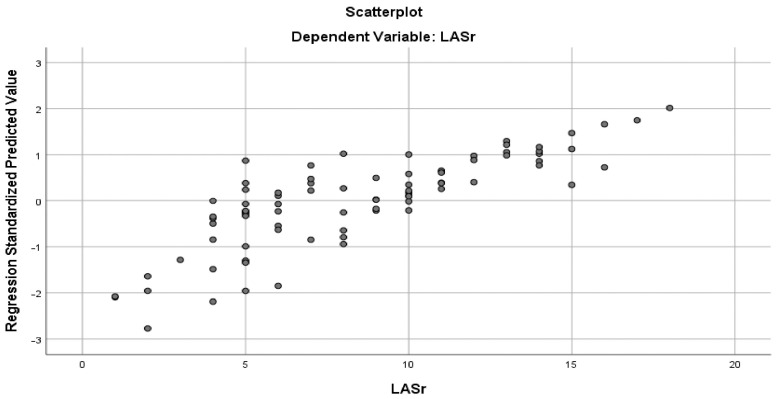
Multiple regression curve showing that PM-AL diameter, TH, and TV were the best predictors of LASr in patients with DCM and the advanced stage of HF.

**Figure 9 biomedicines-13-01753-f009:**
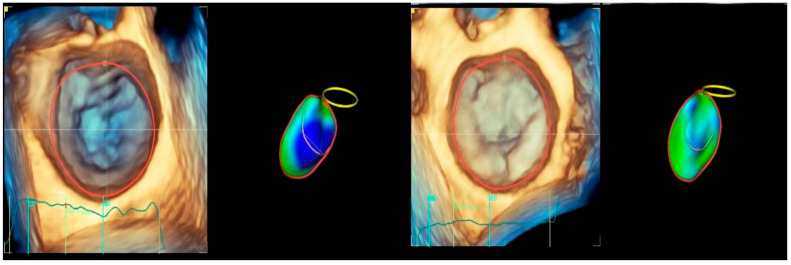
Mitral annulus shape: (**left**) normal subject—saddle shape of mitral annulus; (**right**) patient with chronic left ventricle remodeling—enlarged, flat, and adynamic mitral annulus.

**Figure 10 biomedicines-13-01753-f010:**
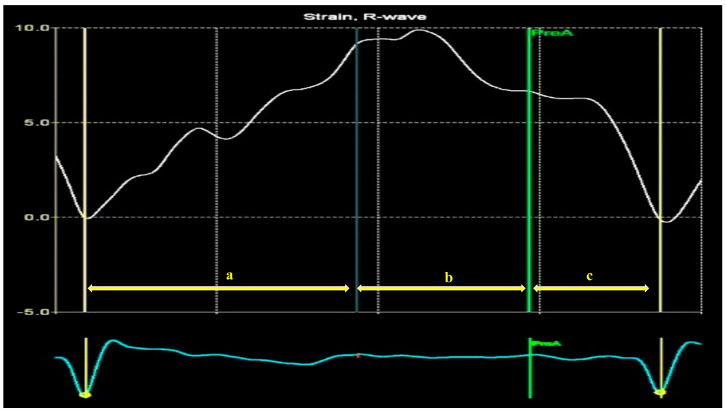
Left atrium strain curve: a—LA reservoir period; b—LA diastasis phase empties; c—LA contraction.

**Table 1 biomedicines-13-01753-t001:** Baseline characteristics in patients with DCM. Data are presented as numbers (n), percentages (%), and means ± standard deviations. Abbreviations: DCM—dilated cardiomyopathy; IDCM—ischemic dilated cardiomyopathy; NIDCM—nonischemic dilated cardiomyopathy; BSA—body surface area; DM—diabetes mellitus; CKD—chronic kidney disease; AF—atrial fibrillation; *p*-value is calculated using the *t*-test for comparison of means.

Parameters	IDCM (38) (46%)	NIDCM (44) (54%)	*p*
Age (years)	63 (±8)	57 (±12)	0.016
Males (%)	23 (60)	28 (64)	0.47
BSA (m^2^)	1.76 (0.15)	1.82 (0.18)	0.1
Hypertension (%)	30 (79)	23 (52)	0.01
DM (%)	23 (61)	16 (36)	0.03
Dyslipidemia (%)	24 (63)	21 (48)	0.004
Obesity (%)	15 (39)	10 (23)	0.1
CKD (%)	18 (47)	12 (27)	0.06
AF (%)	13 (34)	12 (27)	0.5
Deaths	10 (31%)	15 (34%)	0.94
Rehospitalization	17 (45%)	22 (50%)	0.63

**Table 2 biomedicines-13-01753-t002:** Standard echocardiographic parameter measurements in patients with DCM. Data are presented as mean ± standard deviation. Abbreviations: DCM—dilated cardiomyopathy, IDCM—ischemic dilated cardiomyopathy, NIDCM—nonischemic dilated cardiomyopathy, LVEDV—left ventricular end-diastolic volume, LVESV—left ventricular end-systolic volume, LVEF—left ventricular ejection fraction, LAVi—left atrium volume indexed, MR—mitral regurgitation, TR—tricuspid regurgitation, *p*-value is calculated using the *t*-test for comparison of means.

Parameters	IDCM (38) (46%)	NIDCM (44) (54%)	*p*
LVEDV (mL)	203.86 (82.76)	200.97 (71.13)	0.86
LVESV (mL)	149.55 (69.54)	149.61 (63.65)	0.99
LVEF (%)	27.45 (7.7)	26.52 (7.82)	0.59
LAVi (mL/m^2^)	54.23 (12.6)	54.03 (14.84)	0.95
MR ≥ mild (%)	16 (42.1)	17 (38.63)	0.75
TR > mild (%)	10 (26.31)	17 (38.63)	0.24

**Table 3 biomedicines-13-01753-t003:** Parameters resulted from Four-Dimensional Auto Left Atrial Quantification (4D Auto LAQ) and Four-Dimensional Auto Mitral Valve Quantification (4D Auto MVQ). Data are presented as mean ± standard deviation. Abbreviations: IDCM—ischemic dilated cardiomyopathy; NIDCM—nonischemic dilated cardiomyopathy; LASr—left atrium reservoir strain; AA—annular area; AP—annular perimeter; A-P diameter—anteroposterior diameter; PM-AL diameter—posteromedial to anterolateral diameter; CD—commissural distance; Itd—intertrigonal distance; AH—annular height; NPA—nonplanar angle; TH—tenting height; TA—tenting area; TV—tenting volume; *p*-value is calculated using the *t*-test for comparison of means.

Parameters	IDCM (38) (46%)	NIDCM (44) (54%)	*p*
LASr	8.44 (3.68)	8.5 (4.42)	0.94
AA	20.81 (3.91)	19.79 (2.71)	0.17
AP	16.44 (2.05)	16.35 (2.09)	0.84
A-P diameter	4.61 (0.47)	4.56 (0.65)	0.65
PM-AL diameter	5.27(0.72)	5.1 (0.58)	0.24
CD	4.96 (0.64)	4.94 (0.52)	0.81
Itd	3.73 (0.78)	3.67 (0.63)	0.63
AH	6.86 (2.21)	6.5 (2.06)	0.44
NPA	156 (15.7)	155.77 (13.91)	0.83
TH	1.51 (0.31)	1.48 (0.34)	0.68
TA	4.65 (1.25)	4.59 (1.53)	0.84
TV	13.31 (5.32)	11.92 (5.6)	0.27

**Table 4 biomedicines-13-01753-t004:** Pearson correlation between MA parameters and LASr in patients with DCM. Abbreviations: MA—mitral annulus; LASr—left atrium reservoir strain; DCM—dilated cardiomyopathy; AA—annular area; AP—annular perimeter; A-P diameter—anteroposterior diameter; CD—commissural distance; Itd—intertrigonal distance; PM-AL diameter—posteromedial to anterolateral diameter; AH—annular height; NPA—nonplanar angle, TH—tenting height; TA—tenting area; TV—tenting volume; r—coefficient of variation; CI—confidence interval.

Parameter	r	95% CI Interval	*p*
AA	−0.61	−0.73 to −0.45	<0.0001
AP	−0.58	−0.71 to −0.42	<0.0001
A-P diameter	−0.58	−0.71 to −0.42	<0.0001
CD	−0.59	−0.72 to −0.43	<0.0001
Itd	−0.49	−0.64 to −0.31	<0.0001
AH	−0.08	−0.29 to 0.14	0.47
NPA	−0.17	−0.38 to 0.04	0.11
PM-AL diameter	−0.62	−0.73 to −0.46	<0.0001
TH	−0.67	−0.77 to −0.53	<0.0001
TA	−0.66	−0.77 to −0.52	<0.0001
TV	−0.78	−0.85 to −0.67	<0.0001

**Table 5 biomedicines-13-01753-t005:** Univariate logistic analysis for identifying 4D Auto MVQ parameters predictors of 4D LASr in DCM patients. Abbreviations: DCM—dilated cardiomyopathy; LASr—left atrium reservoir strain; AA—annular area; AP—annular perimeter; A-P diameter—anteroposterior diameter; PM-AL diameter—posteromedial to anterolateral diameter; CD—commissural distance; Itd—intertrigonal distance; AH—annular height; NPA—nonplanar angle; TH—tenting height; TA—tenting area; TV—tenting volume; SE—standard error.

Parameters	Unstandardized Coefficients		Standardized Coefficients	t	*p*
	B	SE	Beta		
AA	−0.820	0.119	−0.611	−6.896	0.0001
AP	−1.214	0.190	−0.580	9.032	0.0001
A-P diameter	−4.121	0.647	−0.580	−6.372	0.0001
CD	−4.188	0.639	−0.591	−6.552	0.0001
Itd	−3.080	0.603	−0.496	−5.110	0.0001
PM-AL diameter	−3.815	0.546	−0.616	−6.988	0.0001
AH	−0.153	0.213	−0.080	−0.720	0.473
NPA	−0.051	0.032	−0.178	−1.620	0.109
TH	−8.228	1.029	−0.666	−7.995	0.0001
TA	−1.919	0.243	−0.663	−7.912	0.0001
TV	−0.577	0.052	−0.777	−11.024	0.0001

**Table 6 biomedicines-13-01753-t006:** Stepwise multivariate analysis model in identifying 4D Auto MVQ parameters predictors of 4D LASr in DCM patients. Abbreviations: DCM—dilated cardiomyopathy; LASr—left atrium reservoir: AP—annular perimeter; AP diameter—anteroposterior diameter; PM-AL diameter—posteromedial to anterolateral diameter; TH—tenting height; TV—tenting volume; SE—standard error.

Coefficients ^a^									
Model			Unstandardized Coefficients			Standardized Coefficients		t	*p*
			B	SE		Beta			
1	(Constant)		15.723	0.717				21.940	0.000
	TV		−0.577	0.052		−0.777		−11.024	0.000
2	(Constant)		22.205	2.322				9.563	0.000
	TV		−0.470	0.062		−0.634		−7.611	0.000
	PM-AL diameter		−1.507	0.516		−0.243		−2.921	0.005
3	(Constant)		24.895	2.623				9.490	0.000
	TV		−0.348	0.085		−0.469		−4.104	0.000
	PM-AL diameter		−1.583	0.507		−0.256		−3.124	0.003
	TH		−2.573	1.248		−0.208		−2.061	0.043
**Model Summary ^a^**									
**Model**		**R**	**R Square**		**Adjusted R Square**		**SE of the Estimate**		
1		0.777 ^b^	0.603		0.598		2.583		
2		0.801 ^c^	0.642		0.633		2.469		
3		0.813 ^d^	0.660		0.647		2.420		

^a^ Dependent variable: LASr. ^b^ Predictors: (constant), TV; ^c^ Predictors: (constant), TV, PM-AL diameter; ^d^ Predictors: (constant), TV, PM-AL diameter, TH.

**Table 7 biomedicines-13-01753-t007:** Intra-observer variability of the parameters resulted from 4D Auto LAQ and 4D Auto MVQ in DCM patients. Abbreviations: DCM—dilated cardiomyopathy; AA—annular area; AP—annular perimeter; AP diameter—anteroposterior diameter; PM-AL diameter—posteromedial to anterolateral diameter; CD—commissural distance; Itd—intertrigonal distance; AH—annular height; NPA—nonplanar angle; TH—tenting height; TA—tenting area; TV—tenting volume; CI—confidence interval.

Parameter	Intraclass Coeficient (Average Measures)	95% CI		Value	*p*
		Lower Bound	Upper Bound		
AA	0.961	0.896	0.985	24.539	0.0001
AP	0.985	0.959	0.994	62.145	0.0001
AP diameter	0.941	0.843	0.978	16.422	0.0001
PM-AL diameter	0.966	0.910	0.987	28.379	0.0001
CD	0.961	0.902	0.986	26.381	0.0001
Itd	0.925	0.802	0.972	12.962	0.0001
AH	0.986	0.964	0.995	72.096	0.0001
NPA	0.907	0.756	0.965	10.621	0.0001
TH	0.900	0.529	0.969	15.559	0.0001
TA	0.985	0.953	0.995	84.308	0.0001
TV	0.989	0.970	0.996	82.251	0.0001

**Table 8 biomedicines-13-01753-t008:** Cross-validation procedure results.

Correlations				
Approximately 80% of the Cases (SAMPLE)			Predicted	LASr
0	predicted	Pearson Correlation	1	0.800 *
		Sig. (2-tailed)		0.031
		N	7	7
	LASr	Pearson Correlation	0.800 *	1
		Sig. (2-tailed)	0.031	
		N	7	7
1	predicted	Pearson Correlation	1	0.819 **
		Sig. (2-tailed)		0.000
		N	75	75
	LASr	Pearson Correlation	0.819 **	1
		Sig. (2-tailed)	0.000	
		N	75	75

* Correlation is significant at the 0.05 level (2-tailed). ** Correlation is significant at the 0.01 level (2-tailed).

## Data Availability

Data is contained within the article.

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
