# Peer review of "The Link Between Left Atrial Longitudinal Reservoir Strain and Mitral Annulus Geometry in Patients with Dilated Cardiomyopathy"

_biomedicines, 2025, doi:10.3390/biomedicines13071753_

Round 1

Reviewer 1 Report

Comments and Suggestions for Authors

Dear Authors,

Thank you for submitting your study on the relationship between left atrial reservoir strain (LASr) and mitral annulus (MA) remodeling in dilated cardiomyopathy (DCM). Your use of advanced 4D echocardiography to address this knowledge gap is compelling and provides novel mechanistic insights—particularly how MA geometry parameters (PM-ALd, TH, TV) predict LASr. The rigorous methodology and comprehensive software integration (4D Auto LAQ/MVQ) make this a valuable educational resource for readers seeking to understand advanced imaging techniques.

​​Major Strengths:​​
• The demonstration of interdependence between LA dysfunction and MA remodeling offers important clinical implications for future therapeutic strategies.
• Statistical analyses are appropriately chosen, and complex concepts (e.g., correlation tests) are explained clearly.
• Excellent reproducibility, with strong intra-observer agreement (ICC >0.95).

​​Key Concerns Requiring Attention:​​

​​Data Inconsistencies:​​
The reported body surface area (BSA) of ​​17.64 m² in the IDCM group​​ appears unrealistic. Please verify this value. This is like an elephant to me. 
LASr showed ​​no significant difference​​ between the ischemic (8.44±3.68%) and non-ischemic groups (8.5±4.42%, p=0.94), despite the ischemic patients having higher rates of hypertension, diabetes, and dyslipidemia (p<0.05). This contradicts the expectation that ischemic etiology would worsen LA/MA pathology. Could you discuss possible reasons for this paradox?
Elaborate how LA dysfunction (reduced LASr) mechanically drives MA geometric changes.
LASr is a known prognostic marker—could you analyze its link to clinical outcomes (e.g., death/rehospitalization)?
Report inter-observer variability (beyond intra-observer ICC), though we acknowledge the challenge of sourcing multiple skilled 4D echo analysts.
​​

There are also some minor Revisions:​​
• Update references to include recent LASr prognostic studies (e.g., Hauser et al. EHJ-CVI 2021).

Your findings on LA-MA interplay in DCM hold great educational value. Consider adding a graphical abstract illustrating this interdependence—it would be an excellent teaching tool for students (I plan to share it in my classroom!). Overall, this is promising work that, with minor refinements, will significantly advance our understanding of DCM pathophysiology.

Author Response

The Word Document with the answers was sent.

Reviewer 2 Report

Comments and Suggestions for Authors

Dear colleagues, the importance of new methods to detect pathological changes related to dilated cardiomyopathy is beyond any doubts. Early diagnostics plays a crucial role in successful treatment and can prevent serious complications. The present research covers the most clinically important cases and  provide a simple tool for practitioners that can be effectively used as a part of diagnostics procedures. 

Unfortunately, there are some annoying points reducing the overall scientific soundness of the manuscript:

  1. All the percentages are given with the excessive precision. For example, in table 1:
    IDCM (38) (46.34%)  NIDCM (44) (53.66%). Taking into account the sample size (82 patients) the confidence interval for these percentages is from 36 to 57 % for IDCM and from 42 to 65 % fir NIDCM. Obviously, it does not make any sense to provide the percentages with 2 digits precision. I would recommend rounding all the percentages  down to the integer values. 
  2. The average ages is also given with 2 digits Age IDCM 63 (8.38)  NIDCM 57.43 (11.56). Are you really calculate the age with 3 days step? If so, is such a precision essential for this pathology? There are many values with the same issue of correct rounding, e.g. table 4 "Pearson correlation coefficients..." . I would recommend editing these values throughout the manuscript.
  3. The proposed multiple regression model (figure 8) demonstrates quite a moderate performance in R squared metrics. From practical point of view it means that it can not help too much in drawing diagnostic conclusions. I would recommend measuring the model performance using more metrics and cross validation procedure.

Author Response

The answer is in the Word Document attached.

Reviewer 3 Report

Comments and Suggestions for Authors

Dear Authors,

The study conducted to explore the association between left atrial longitudinal reservoir strain and mitral annular geometry in patients with dilated cardiomyopathy provides valuable insights for both researchers and general physicians. The study is well-designed and has been executed commendably. However, a few minor revisions are recommended to enhance the clarity and overall quality of the manuscript:

1. The manuscript mentions that case histories of 97 heart failure (HF) patients were collected. Since some of these patients may have had comorbid conditions such as diabetes or hypertension, which could potentially influence the heart failure data, please clarify how such associated conditions were addressed or accounted for in your analysis.

2. The exclusion criteria for patient selection are not clearly described. Kindly elaborate on these criteria to ensure better transparency and understanding of the study design.

3. Please consider rewriting the conclusion to succinctly highlight the key findings and the clinical significance of your study outcomes.

Comments on the Quality of English Language

The quality of the language need improvement.  Please avoid personal pronouns like we, our etc.

Author Response

(The authors gave the same response as above.)

Reviewer 4 Report

Comments and Suggestions for Authors

The submitted manuscript, titled “The link between left atrial longitudinal reservoir strain and mitral annulus geometry in patients with dilated cardiomyopathy" shows potential for publication in this journal. This manuscript addresses an important and underexplored area in cardiovascular imaging, specifically the relationship between left atrial reservoir strain (LASr) and mitral annulus (MA) geometry in patients with dilated cardiomyopathy (DCM), using advanced four-dimensional transthoracic echocardiography (4D TTE) techniques. The study is timely and relevant for clinicians and researchers interested in structural heart disease, echocardiographic techniques, and heart failure prognostication. However, the manuscript would benefit from refinement in language, structure, and interpretation of results. A few methodological limitations and reporting issues also need to be addressed. Below are comments and suggestions for revision as major.

Please address the following comments:

  • The introduction could better emphasize why understanding the link between LASr and MA geometry is clinically significant, especially in the context of heart failure management.

  • Please specify the vendor and version of the echocardiographic software earlier in the Methods.

  • Details on blinding or interobserver variability would strengthen the validity of image analysis.

  • Table 4 (Pearson correlations): Indicate whether multicollinearity was assessed before regression analysis.

  • The Discussion is well-referenced and thorough but overly long. It would benefit from clearer subheadings and a more structured approach (e.g: principal findings, comparison to previous work, clinical relevance).

  • The potential influence of atrial fibrillation on LASr should be discussed further, given its presence in ~30% of the cohort.

  • The conclusion aligns with the findings but should include more specific clinical applications and future research directions.

Comments on the Quality of English Language
  • The manuscript requires extensive language editing for grammar, punctuation, and sentence structure. Several sentences are overly long and difficult to follow. Consider breaking complex sentences into shorter, clearer ones.
  • Consistency in terminology (e.g., “mitral annulus” vs “MA”, “left atrial strain” vs “LAS”) should be maintained.

Author Response

(The authors gave the same response as above.)

Round 2

Reviewer 4 Report

Comments and Suggestions for Authors

The submitted manuscript, titled “The link between left atrial longitudinal reservoir strain and mitral annulus geometry in patients with dilated cardiomyopathy" shows potential for publication in this journal. The improvements from the previous version are commendable, particularly the enriched methodology and detailed analysis.

However, several issues primarily related to language clarity, manuscript structure, and result interpretation need further attention before the manuscript can be considered for publication. Below are comments and suggestions for revision as major.

  • The manuscript requires thorough professional English language editing. Numerous grammatical issues, typographical errors, and awkward phrasing detract from readability.

  • Although comparisons between ischemic and nonischemic DCM were performed, the manuscript should emphasize that most differences were not statistically significant. Highlight the predictive significance of PM-AL diameter, TH, and TV in relation to LASr more clearly in both results and discussion.

  • The study would benefit from a clearer discussion on how these findings can be translated to clinical practice for example, their potential in risk stratification or surgical planning for DCM patients.

  • The discussion is excessively long and repetitive, often reiterating background knowledge rather than focusing on study-specific insights. Please streamline and focus on interpretation.

Comments on the Quality of English Language

1) The abstract is dense. Consider reducing redundancy and highlight only the key methodology and statistically significant findings particularly the predictive value of PM-AL diameter, TH, and TV.

2) The discussion is excessively long and repetitive, often reiterating background knowledge rather than focusing on study-specific insights. Please streamline and focus on interpretation.

Author Response

The abstract was modified.

The discussions were reviewed.

Round 3

Reviewer 4 Report

Comments and Suggestions for Authors

Dear Editor,

Thank you for the opportunity to review this manuscript. I sincerely appreciate the trust you have placed in me. Please find my review comments below.

The submitted manuscript, titled “The link between left atrial longitudinal reservoir strain and mitral annulus geometry in patients with dilated cardiomyopathy" shows potential for publication in this journal. This version reflects a well-conducted, clinically valuable study that demonstrates the utility of 4D echocardiographic metrics in assessing structural remodeling in DCM. Advising professional language editing before publication.

Comments on the Quality of English Language

Advising professional language editing before publication. 

Obaserved inconsistent phrasing, and overly long sentences (especially in the Discussion).

Author Response

Dear Sirs,

I would like to inform you that the manuscript has been evaluated by an authorized English speaker. 

I do not see any reason for using the "Language editing" service, as the price is too high. 

Two Reviewers agree with this form of the manuscript. 

Thank you very much for understanding,

Dr Despina Toader